# A Primitive Trait in Two Breeds of *Equus Caballus* Revealed by Comparative Anatomy of the Distal Limb

**DOI:** 10.3390/ani9060355

**Published:** 2019-06-14

**Authors:** Sharon May-Davis, Zefanja Vermeulen, Wendy Y. Brown

**Affiliations:** 1Canine and Equine Research Group, University of New England, Armidale, NSW 2351, Australia; wbrown@une.edu.au; 2Equine Studies, 41157 LN Asch, The Netherlands; info@equinestudies.nl

**Keywords:** interosseous muscle II, III and IV, Dutch Konik, Bosnian Mountain Horse, Donkey, Przewalski’s horse, Zebra, Atavism

## Abstract

**Simple Summary:**

Understanding the complexities and evolutionary links between extinct and extant equids has been vital to genetic conservation and preservation of primitive traits. As domestication of the equid expanded, the loss of primitive traits that ensured survival in a wild environment has not been documented. In this study, the presence of functional interosseous muscle II and IV in the distal limb has been reported, and yet its existence could only be confirmed in relatives and two closely bred descendants of the extinct Tarpan. The morphology described was ligamentous in structure displaying clear longitudinal fibres with a skeletal origin and soft tissue insertion into the medial and lateral branches of the interosseous muscle III (suspensory ligament) dorsal to the sesamoids, similar in orientation to the flexor digitorum profundus ligamentum accessorium (inferior check ligament). Hence, providing a functional medial and lateral stability to the metacarpophalangeal joint (fetlock joint), which equates to one of the functions of the medial and lateral digits in the *Mesohippus* and *Merychippus.* The comparable anatomic links between species of the same family that experienced geographical isolation yet display structural conformity appears to be in response to a specific environment. Surmising this potential remnant of functional evolution is a primitive trait and not a breed anomaly.

**Abstract:**

The 55-million-year history of equine phylogeny has been well-documented from the skeletal record; however, this is less true for the soft tissue structures that are now vestigial in modern horse. A recent study reported that two ligamentous structures resembling functional interosseous muscle II and IV were evident in Dutch Konik horses. The current study investigates this finding and compares it to members of the genus *Equus* to identify either a breed anomaly or functional primitive trait. Distal limbs (*n* = 574) were dissected from four species of *Equus*; *E. caballus*, *E. asinus*, *E. przewalskii* and *E. quagga boehmi*. *E. caballus* is represented by 18 breeds of horse, including the primitive Dutch Konik’. The interosseous muscle II and IV were evident in all four species, but only two breeds of *E. caballus* expressed this trait-the Dutch Konik and Bosnian Mountain Horse. These two breeds were the only close descendants of the extinct *Equus ferus ferus* (Tarpan) represented in this study. In conclusion, the interosseous muscle II and IV originated from the distal nodule of metacarpal II and IV, respectively, and inserted into the corresponding branches of interosseous muscle III proximal to the sesamoids. This suggests a functional role in medial and lateral joint stability and a primitive trait in modern equids.

## 1. Introduction

Aristotle (384–322BC) is regarded by many as the founder of comparative anatomy because of his methodical and systematic approach to the study of animals [1]. His rigorous and comprehensive methods provided the basis for numerous original theories, including Charles Darwin’s 1859 publication ‘On the Origin of Species’ [1,2]. Aristotle believed that ‘form and function’ were integral parts of the same science and his in-depth knowledge of bodily systems was likely derived from direct observations and dissections [1]. As medical practitioners began to embrace post mortem instruction, medieval practices were abandoned and the modernisation of medicine as we know it today was founded [3].

By the 1800s, this new anatomic enlightenment inspired many scholars and scientists, whilst simultaneously invoking strong academic debate and controversy [4,5]. Nonetheless, the science of comparative anatomy prevailed and shaped the fundamental principles of taxonomy in extinct and extant species [6]. Even when geographically isolated, the similarity between species was undeniable, and furthermore, it became evident that structural conformity was in direct response to the environment [2,6]. These distinctions were developed through the study of comparative anatomy and formed the basis of sound phylogenetic interpretation and evolutionary taxonomy [6].

It was this methodical approach to comparative observation that palaeontologist O. C. Marsh utilised when he correlated fossil evidence and established the 55-million-year history of equine phylogeny [7]. His research deciphered the skeletal transformation from polydactyl to monodactyl, including connective soft tissue structures operating the distal digits [7,8]. These skeletal archives presented a convoluted, but connected history, originating with the polydactyl *Eohippus*, until the present-day monodactyl *E. caballus* [7,8,9]. The adaptations of the distal digits through the millennia corresponded to a period in history when the climate became progressively drier and open plains expanded at the expense of forests [8].

For the polydactyl horse, living in an open environment exposed it to greater stressors, namely foraging competition and predative pressures; hence, speed and endurance became a vital trait in order to survive. However, as its distal limb was designed for stability and not speed, the mass of the extra digits became an energetic cost that hindered locomotive efficiency [8,10,11]. Consequently, the distal limb required new adaptions that favoured efficient fore and aft linear movement with less flexibility [8,11]. Hence, in direct response to its environment, the polydactyls distal limb evolved by favouring the reduction of the medial and lateral digits, whilst elongating the middle; thus, becoming the tight jointed, rigid hoofed monodactyl that we encounter today [7,8,9,10,11].

In the modern horse, polydactyl atavism or primitive characteristics have been reported in the appendicular skeleton and although considered rare, as have reports of atavistic or vestigial soft tissue structures in the distal limb. [12,13,14,15]. Evidently, these soft tissue structures were once strong and functional in the polydactyl, but now, the morphology is regarded as rudimentary and ineffectual (Figure 1) [12,15]. However, two atavistic soft tissue structures reported in the forelegs; the medial or II interosseous muscle (IM2) and lateral or IV interosseous muscle (IM4) are of interest to this study. Both are vestigial remnants of medial and lateral limb reduction, but in the modern horse, there is only one interosseous muscle considered functional—the middle or interosseous muscle III (IM3) [12,13,14,15,16]. This muscle (IM3) evolved even further and now contains a large number of strong ligamentous fibres with elastic properties that functionally support the metacarpophalangeal joint (fetlock joint) when the limb is either standing, or during locomotion [15,16].

When present in the modern horse, the IM2 and IM4, have been described as thin, pale and fleshy ligamentous structures originating from the distal nodule of the metacarpal II and IV that ends inconspicuously near the metacarpophalangeal joint (metacarpal III, phalanx I and 2 sesamoids) [12,15]. In contrast, a recent study revealed strong chord-like bands in primitive Dutch Konik horses originating from the distal nodules of the metacarpals II and IV, and metatarsals II and IV [18].

Therefore, the aim of this study is to investigate the strong chord-like bands reported in the primitive Dutch Konik horse and compare them to domestic horse breeds along with other available species in the genus *Equus*. We describe the morphology, anatomic origin and insertion, and postulate the function of the bands, with the objective to provide a better understanding of whether they are a rare finding, a breed anomaly or a primitive trait. We conclude the presence of an IM2 and or IM4 in its current morphological form as noted in this study relates to a primitive trait, and furthermore, one that is possibly a functional remnant of limb reduction in response to an undulating environment.

## 2. Materials and Methods

### 2.1. Ethical Statement

No equids were euthanized for the purpose of this study. All observations were obtained post mortem from deceased equids sourced through abattoirs; government culling programs; designated euthanasia for humane purposes or via natural causes, e.g., environmental, captive domestication or zoos.

### 2.2. Animal Details

Dissections were performed on 575 distal limbs (DL) from 151 individual equids of the genus *Equus*; 488 DL were sourced from 122 *E. caballus* (domestic and primitive); 15 DL from 9 *E. przewaslkii* (Przewalski’s horse); 11 DL from 3 *E. asinus* (donkey) and 4 DL from 1 *E. quagga boehmi* (Grant’s zebra). Animals ranged in age from 2 days to 30 + years.

The 488 DL from *E.caballus* were sourced in Australia (308); The Netherlands (64); Japan (40); New Zealand (36); United Kingdom (32); Sweden (4) and Slovenia (4). The 15 DL from *E. przewaslkii* were sourced from Hungary (Budapest Zoo and Hortobagy National Park); the 11 DL from *E. asinus* were sourced from Tennant Creek Station, Northern Territory, Australia and the 4 DL from *E. quagga boehmi* were sourced from Wildlands Adventure Zoo Emmen, The Netherlands. Eighteen designated horse breeds were represented in *E. caballus* [19]: Thoroughbred (208 DL); primitive Dutch Konik (DK) (61 DL); Crossbreds (56 DL); Warmbloods (56 DL); Australian Stock Horse (52 DL); Standardbred (16 DL); Quarter Horse (16 DL); Welsh Mountain Pony (12 DL); Arabian (eight DL); Irish Sport Horse (eight DL); Appaloosa (eight DL); Hunter (eight DL); Hackney (eight DL); Exmoor Pony (eight DL); Fjord (eight DL); Icelandic (four DL), Morgan (four DL) and Bosnian Mountain Horse (BHM) (four DL).

The primitive Dutch Konik horses provided 61 DL that were sourced from five unrelated herds located in nature reserves across The Netherlands; De Rug (eight); Geuzonbos (10); Loevestein (12); Leeuwin (31).

There were 291 forelegs and 284 hindlegs. Each foreleg displayed a metacarpal II and metacarpal IV (MC2 and MC4 respectively). Each hind leg displayed a metatarsal II and metatarsal IV (MT2 and MT4 respectively). In total 1146 MC2, MC4, MT2 and MT4 were examined with associative soft tissue structures for this study (interosseous muscles II, III and IV; IM2, IM3 and IM4 respectively).

Note: Not all four DL were available per individual equid and the DL ranged from one to four per individual in this study.

### 2.3. Dissections

The forelegs and hindlegs were skinned from the distal radius and distal tibia to the distal phalanx II (Ph. II). Disarticulation of the distal limbs occurred at the articulation of the first and second carpal rows in the carpus of the foreleg, and the articulation of the tibia and talus tarsal bone in the tarsus of the hindleg. Thus, severing all connective soft tissue structures and releasing the distal limb for closer examination. Resection of the flexor tendons (flexor digitorum superficialis and flexor digitorum profundus, or SDFT and DDFT, respectively), the flexor digitorum profundus ligamentum accessorium or inferior check ligament (ICL), nerves and various vessels from the palmer aspect of the MC3 to phalanx II exposes the greater part of MC2, MC4 and IM3 (Figure 2).

Careful resection of the IM3 from its origin on the proximal palmar surface of MC3 to the level of the distal nodules of the MC2 and MC4 reveals, if present, the IM2 and IM4 originating from the distal nodules of MC2 and MC4. When not present the resection of the IM3 and its medial and lateral branches continued distally until insertion onto the respective medial and lateral sesamoids. However, when the IM2 and IM4 were present, the resection required slow and precise strokes as to not compromise the origin or insertion of the IM2 and IM4. Any connective fascia was then carefully removed revealing the IM2 and IM4 in its entirety. This process of resection was repeated for the DL in each hindleg along its plantar surface.

## 3. Results

All equids expressed the IM3, however the IM2 and IM4 were noted in 2/18 breeds of *E. caballus* (DK and BMH), 9/9 *E. przewalskii* (Przewalski’s horse), 3/3 *E. asinus* (donkeys) with *E. quagga boehmi* (Grant’s zebra) expressing the IM4 in the forelegs only (Table 1).

Anatomically, the IM2 and IM4 originated from the nodules of the MC2 and MC4, and the MT2 and MT4, respectively. The point of insertion for the IM2 was the medial branch of the IM3; the point of insertion for the IM4 was the lateral branch of the IM3. All points of insertion were dorsal to the sesamoids. The thickness of the IM2 and IM4 appeared to remain constant from origin to insertion (Figure 3a–c). At the point of insertion, the fibres of the IM2 and IM4 appeared to interconnect with those of the IM3. These fibres were longitudinal in arrangement from origin to insertion and appeared consistent with a collagenous protein with elastic properties, such as that found in tendons or special ligamentous structures such as the IM3.

## 4. Discussion

In this study, the presence of a strong cord-like band resembling a tendon or ligament in the distal limb was identified and described in four species of the genus *Equus*. The bands originated from the distal nodules of MC2, MC4, MT2 and MT4, corresponding to the previously reported description of the atavistic IM2 and IM4 [12,15]. However, in contrast to a thin and feeble ligamentous structure, morphological variations in size and insertion were noted, with conspicuous cord-like bands inserting into corresponding branches of the IM3 (Figure 3a–c). The described bands were observed in one domestic breed (BMH) out of 18 in *E. caballus*; one primitive breed (DK) out of one in *E. caballus*; three out of three in *E. asinus* (donkeys); nine out of nine in *E. przewalskii* (Przewalski’s horse) and one out of one in a *E. quagga boehmi* (Grant’s zebra).

Of the 18 breeds representing *E. caballus*, only the close descendants of the extinct *E. ferus ferus* (Tarpan), the DK and BMH expressed the IM2 and IM4 [21,22]. This coincides with a previous study, where the DK and BMH were the only two out of the 20 breeds of *E. caballus* that exhibited a full nuchal ligament lamellae [23]. Thus, implying the ligamentous structures noted in the current and previous study, may well be attributed to *E. ferus ferus* (Tarpan) as a heritable characteristic [23].

The strong cord-like bands described in this study were not vestigial, but displayed clear demarcations associated with origins and insertions as would be expected of a functional structure. This is somewhat similar to the ICL that originates from the palmar surface of the distal row of carpals in the carpus and inserts into the DDFT providing stability and support [12,15]. In fact, both structures have proximal skeletal origins with distal soft tissue insertions and identical orientation, implying similar function. Furthermore, the morphology of the strong bands described in this study, present surprising similarities within members of the genus *Equus* that are geographically isolated, suggesting heritably from a common ancestor.

As previously described, vestigial or atavistic soft tissue structures in the distal limb are reported as rudimentary and non-functional. However, identifying new soft tissue structures that are functional in an equid’s distal limb, is not unprecedented [12,15,24,25]. A recent study in a miniature donkey (*E.asinus*), introduced a new ligament on the palmar surface of metacarpal III with functional capabilities of stabilising the SDFT [24]. An earlier study reported fascial bundles in five horses descending distally from the nodule of the splint bone and attaching to the third metacarpal, securing the smaller metacarpal to the larger one [25]. Although reports of soft tissue structures like those described in this study have not been found for extant members of *Equus*, this does not apply to families closely related to *Equidae* or polydactyls. In fact, all three interosseous muscles have been reported in extant and extinct polydactyls [24,25,26,27,28,29].

Studies have identified the presence of the IM2, IM3 and IM4 in the *Hippopotamus amphibious*, *Acrocodia indica* (Malayan tapir) and members of the genus *Caninis*, including *Marsupialias* such as the *Thylacinus cynocephalus* (thylacine), of which the origins and insertions displayed similar morphology in the *Hippopotamus amphibious* and *Acrocodia indica* to those found in this study [26,27,29]. The IM3 and IM4 were noted in *Suidaes* (pigs), *Ovis aries* (sheep) and *Lama glama* (llamas), but not *Bos taurus* (cattle) or *Camelids*, where only the IM3 was mentioned per digit [26,30,31,32,33,34]. There were no definite anatomic origins or insertions noted for *Suidaes* (pigs) or *Ovis aries* (sheep); however, the *Lama glama* (llamas) presentation corresponded anatomically to the current study [26,30,31].

Many of the species previously mentioned belong to the unguligrade mammals within the orders perissodactyl and cetartiodactyl [33]. The latter order provides evidence of skeletal entheses patterns in the distal limb, where the IM3 and IM4 attached in an extinct species of palaeomerycid, a deer-like creature from the Miocene epoch [35,36,37]. During the same epoch, the perissodactyl forebears of *Equidae*, the tridactyl *Mesohippus* and *Merychippus* had anatomic structures comparable to the *Acrocodia indica* (Malayan tapir), including interosseous muscles [38]. The three digits in the tridactyl were referred to as II, III and IV (medial, middle and lateral respectively), with III being the largest, widest and dominant during weightbearing [38,39]. Each interosseous muscle corresponded to its metacarpal or metatarsal; hence, the interosseous muscles in the *Mesohippus* and *Merychippus* would be labelled IM2, IM3 and IM4 from which they originated [12,15]. In current literature, this anatomical arrangement remains constant between species, whether extinct or extant.

Furthermore, it has been suggested that functionally the medial and lateral digits in prehistoric tridactyls assisted in the prevention of lateral dislocation of the fetlock joint while increasing agility and maneuverability, whereas in *Merychippus*, these digits also helped increase traction over soft ground and savannas [38]. This concurs with the adaptive responses that feature in the *Camelid’s* specialized distal limb to its sandy environment [32]; this applies to *E. asinus* (donkeys), where specific adaptations were in direct response to a specific environment, which has already been reported in ungulate morphology [2,6,40]. Unlike domestic *E. caballus*, the distal limb in the *E. asinus* (donkey) ends in a small boxy upright hoof with thick outer walls that are extremely strong and pliable, permitting greater accuracy during placement in rocky and difficult mountainous terrain [40,41].

Looking into the phylogeny of *Equidae*, the genus *Equus* emerged from the hippomorphs 3.8 million years ago (Mya); *Equus asinus* (donkey) diverged from its common ancestor with the caballine horses 2.1 Mya; *Equus quagga*—the zebras 1.2–1.6 Mya and *E. przewalskii* 50,000 years ago [36]. In this study, all three genera presented with the IM2 and or IM4, which suggests the strong cord-like bands have a functional role in their environment, as per the ancestral horse *Mesohippus* and *Merychippus* [38]. This also equates to the DK, BMH, Przewalski’s horse, *E. asinus* (donkey) and *E. quagga* (zebra) that have evolutionary pathways involving mountainous terrain, soft pliable surfaces and more specifically, undulating environments where medial and lateral stability of the metacarpalphalangeal joint (fetlock joint) is necessary [21,22,36,42]. Therefore, it could be postulated that a functional IM2 and IM4 are primitive traits in *E. caballus*.

## 5. Conclusions

The IM2 and IM4 in modern *Equus* have been described as vestigial and non-functional remnants from tridactyl forebears; where it was postulated, they were structurally functional for traction and stability in the ancient equid. With distant relatives and four genera of *Equus* displaying the IM2 and IM4, we conclude the anatomic trait as described in this study is not a breed anomaly, but a primitive trait found in two breeds of *E. caballus* known as the DK and BMH.

## Figures and Tables

**Figure 1 animals-09-00355-f001:**
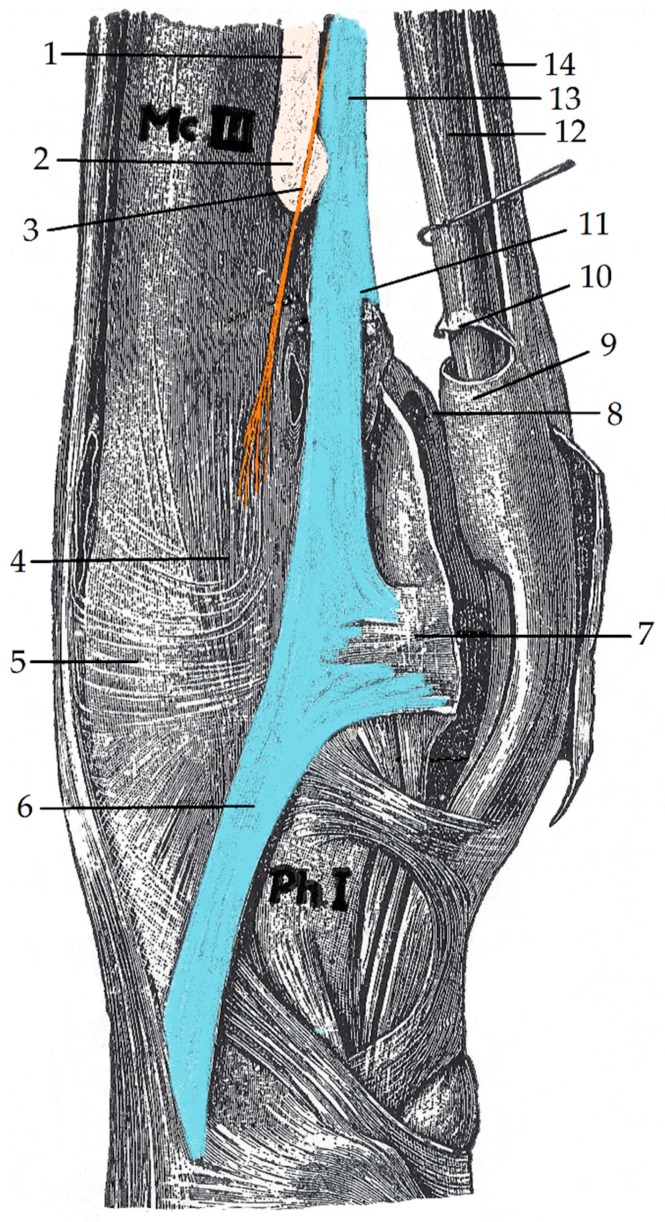
The medial view of the metacarpophalangeal joint in the equine right foreleg. The metacarpal II and its distal nodule (beige) and rudimentary tendon of the interosseous muscle II (orange) are noted. The medial branch of the interosseus III muscle, its bifurcation into medial and lateral branches of the interosseus III muscle have been depicted (blue). Adapted from Schmaltz 1911 [17]. Mc.III–metacarpal III; Ph.I–phalanx I; 1. metacarpal II (beige); 2. distal nodule metacarpal IV (beige); 3. rudimentary tendon of the interosseous muscle II (orange); 4. Medial collateral ligament of the metacarpophalangeal joint; 5. anterior fascia of the metacarpophalangeal joint; 6. medial branch of the interosseus III muscle (blue); 7. medial aspect of the collateral sesamoidean ligament; 8. intersesamoidean ligament; 9. manica flexorium; 10. proximal aspect of the digital tendon sheath; 11. bifurcation of the branches of the interosseus III muscle; 12. deep digital flexor tendon (DDFT); 13. interosseus III muscle; 14. superficial digital flexor tendon (SDFT).

**Figure 2 animals-09-00355-f002:**
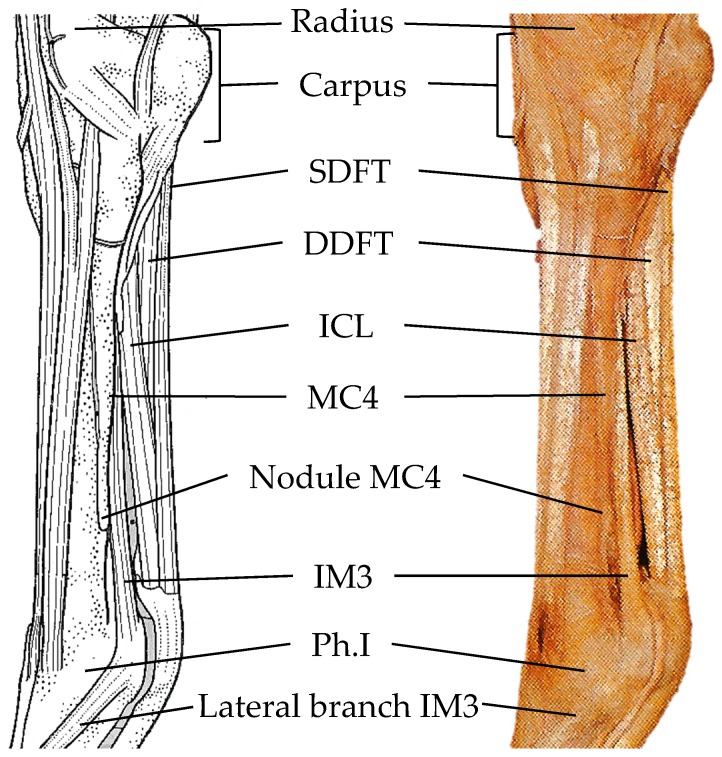
Left foreleg: the lateral view of the carpus, metacarpal III, metacarpophalangeal joint and proximal phalanx I with major soft tissue structures. Adapted from Ashdown and Done [20]. SDFT: flexor digitorum superficialis, DDFT: flexor digitorum profundus, ICL: profundus ligamentum accessorium or inferior check ligament, MC4: metacarpal IV, IM3: interosseous muscle III.

**Figure 3 animals-09-00355-f003:**
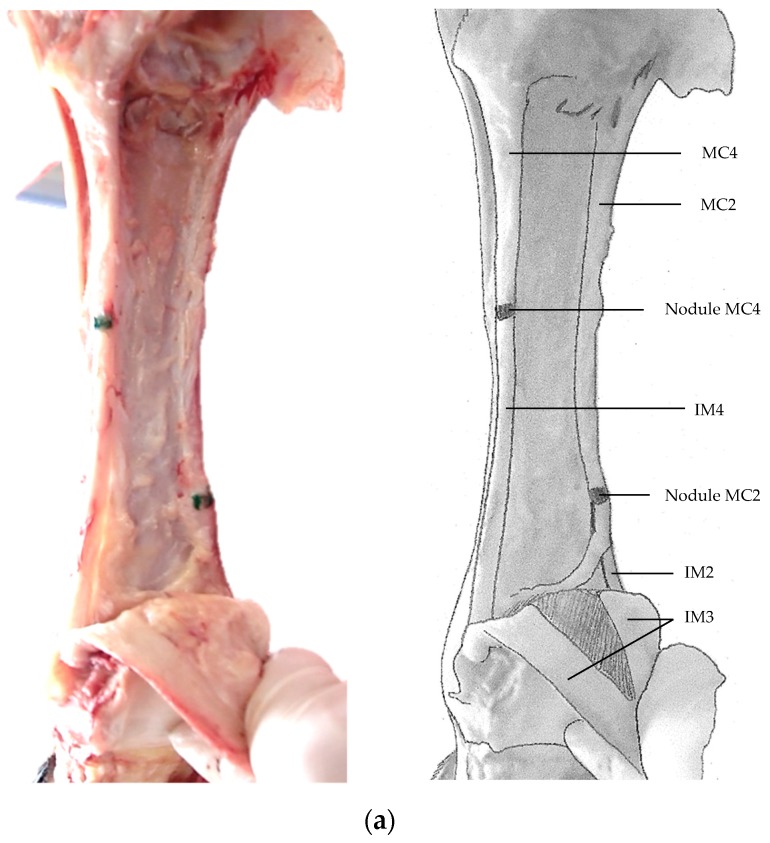
Resection of the distal foreleg revealing IM2 and IM4. (**a**). Left foreleg: the palmar view from the carpus to proximal Ph.I in a seven year old female *E. caballus* (Dutch Konik) from De Rug, The Netherlands, displaying the interosseous muscles IM2 and IM4, and resected IM3. (**b**). Left foreleg: the palmar view from the carpus to proximal Ph.I in a 11 year old male *E. przewalskii* (Przewalski’s horse) from Hortobagy National Park, Hungary, displaying interosseous muscles IM2 and IM4, and the resected IM3. (**c**). Right foreleg: the lateral palmar view from the carpus to proximal Ph.I in a 23 year old female *E. quagga boehmi* (Grant’s zebra) from Wildlands Adventure Zoo Emmen, The Netherlands, displaying interosseous muscle IM4 and the resected IM3 (IM2 was not present). MC2: metacarpal II, MC4: metacarpal IV, IM3: interosseous muscle III, IM4: interosseous muscle IV.

**Table 1 animals-09-00355-t001:** The documented occurrence of interosseous muscles II and IV (IM2 and IM4) in the distal limbs (*n* = 575) of four *Equus* species, dissected postmortem.

Species	Column Title	Distal Limbs (n) ^1^		IM2 (n)		IM4 (n)	
		Left	Right	Left	Right	Left	Right
*Equus caballus*: Domestic horse	Fore	121	121	1 *	1 *	1 *	1 *
(17 breeds *n* = 121; DL *n* = 484)	Hind	121	121	1 *	1 *	1 *	1 *
*Equus caballus*: Primitive horse	Fore	15	14	13	12	13	12
(Dutch Konik *n* = 17; DL = 61)	Hind	15	17	15	15	15	16
*Equus przewalskii*	Fore	9	2	9	2	9	2
(Przewalski’s horse *n* = 9; DL = 15)	Hind	2	2	2	2	2	2
*Equus asinus*	Fore	3	3	3	3	3	3
(Donkey *n* = 3; DL = 11)	Hind	3	2	3	2	2	2
*Equus quagga boehmi*	Fore	1	1	0	0	1	1
(Grant’s zebra *n* = 1; DL = 4)	Hind	1	1	0	0	0	0

^1^ Note: Not all four limbs were available for some animals. * Denotes Bosnian Mountain Horse. IM2: Interosseous muscle II, IM4: Interosseous muscle IV.

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
