# Peer review of "A Primitive Trait in Two Breeds of Equus Caballus Revealed by Comparative Anatomy of the Distal Limb"

_animals, 2019, doi:10.3390/ani9060355_

Round 1

Reviewer 1 Report

Dear Authors,

all the comments are in form of 'sticky notes'

in the pdf attached.

Author Response

Dear Reviewer,

Thank you for taking the time in reviewing this manuscript. Your points have been noted and changes made accordingly.

The title recommendation has been altered with a small addition included.

Diagrams / drawings and photographs have been adjusted for further clarification.

In Material and Methods: all equids were euthanised for various reasons for which we were not privy to every detail. However, I hope the overview in the first paragraph meets with your approval.

I have made a significant number of NAV alterations; however, my concern is that most folks involved with day-day horse affairs or conservation matters will struggle with the terminology. Hence, I have incorporated both where possible to best meet the audience that this paper will reach.

Kind regards,

Sharon May-Davis

Reviewer 2 Report

The authors submitted a manuscript claiming for a primitive anatomical trait of equine distal anatomy. However, I am suspicious about such qood news. I think that the current illustration is not helpful. It would be better, if you could place drawings adjacent to the photographs, in order to offer a more clear picture of the trait. The table and the numbers are correct? It would be useful if you could adopt the terminology of the Nomina Anatomica Veterinaria regarding the anatomical parts(bones, ligaments, muscles), and if could clarify and implement correctly  the terms, species, breed, genera e.t.c. 

The authors must clarify the zoological classification of the specimens. I am not sure if the thoroughbred is a distinct breed e.t.c. The authors must adopt the terminology of the Nomina Anatomica Veterinaria to describe the various anatomical structures. In the Figure 4, I can not understand what is illustrated, since the photo is focused at a very limited point. In the Figure 3, the authors do not provide information about the limb. Which distal limb? Is it fore or hind limb? Is it right or left? What is the view of each picture? Is it caudal, lateral? Regarding the Figure 2, I would suggest to consult and use other drawings which illustrate more detailed that region. Since the structure of that anatomical region is complicated. I think that a combination of drawings and photos could better illustrate the “trait”. In the Table 1, the number of left fore distal limbs in Equus przewalskii is 9 but the number of left fore IM2 is 11. Is it correct? I believe that apart of the revision that must be performed, a different look at the manuscript should be taken.

Author Response

Dear Reviewer,

Thank you for taking the time in reviewing this manuscript. Your points have been noted and changes made accordingly.

Diagrams / drawings and photographs have been adjusted for further clarification.

Table 1. has been corrected.

I have made a significant number of NAV and species alterations; however, my concern is that most folks involved with day-day horse affairs or conservation matters will struggle with the terminology. Hence, I have incorporated both where possible to best meet the audience that this paper will reach.

Thoroughbred’s etc., are referred to as a breed, so to clarify this further, I have referenced a highly reviewed book on horse breeds.

I hope this meets with your requirements.

Kind regards,

Sharon May-Davis

Round 2

Reviewer 2 Report

Please take a look at the "Ethical statement" paragraph.